# Projection neurons in *Drosophila* antennal lobes signal the acceleration of odor concentrations

Anmo J Kim[†], Aurel A Lazar*, Yevgeniy B Slutskiy

Department of Electrical Engineering, Columbia University, New York, United States

**Abstract** Temporal experience of odor gradients is important in spatial orientation of animals. The fruit fly *Drosophila melanogaster* exhibits robust odor-guided behaviors in an odor gradient field. In order to investigate how early olfactory circuits process temporal variation of olfactory stimuli, we subjected flies to precisely defined odor concentration waveforms and examined spike patterns of olfactory sensory neurons (OSNs) and projection neurons (PNs). We found a significant temporal transformation between OSN and PN spike patterns, manifested by the PN output strongly signaling the OSN spike rate and its rate of change. A simple two-dimensional model admitting the OSN spike rate and its rate of change as inputs closely predicted the PN output. When cascaded with the rate-of-change encoding by OSNs, PNs primarily signal the acceleration and the rate of change of dynamic odor stimuli to higher brain centers, thereby enabling animals to reliably respond to the onsets of odor concentrations.

*For correspondence: aurel@ee. columbia.edu

**Present address:** [†]The Rockefeller University, New York, United States

**Competing interests:** The authors declare that no competing interests exist.

## Introduction

Odor distribution in nature is intermittent and dynamic (*Murlis et al., 1992*; *Vickers et al., 2001*), and animals have evolved the ability to detect and respond to temporal variation of odor stimuli (*David et al., 1983*; *Thesen et al., 1993*; *Vickers et al., 2001*; *Porter et al., 2007*; *Semmelhack and Wang, 2009*; *Kato et al., 2014*). In one of the most sophisticated examples, *Drosophila* larvae with only a single functional olfactory sensory neuron (OSN) are capable of moving toward a droplet of an attractive odor by actively orienting themselves (*Louis et al., 2008*). Similarly, adult fruit flies exhibit robust odor-guided behaviors such as turning upwind in flight upon contact with an attractive odor plume (*Budick and Dickinson, 2006*) and staying within a specific odor zone (*Semmelhack and Wang, 2009*). In order to enable such odor-guided tasks, it is essential for any olfactory system to process time-varying features of olfactory stimuli and supply behaviorally relevant information to higher brain centers.

Several recent studies have investigated how dynamic olfactory stimuli are processed in insect early olfactory systems (systems consisting principally of OSNs and projection neurons [PNs]) and observed significant temporal processing of odor signals (*Bhandawat et al., 2007*; *Geffen et al., 2009*; *Kim et al., 2011*; *Nagel and Wilson, 2011*; *Martelli et al., 2013*). Most of these studies employed a simple odor delivery system that generated step-pulse-like odor stimuli without directly monitoring the actual odor concentration levels. For a rigorous understanding of sensory processing, however, it is essential to precisely measure the input stimuli and systematically explore the input space, as has been successfully done in the field of vision and audition (*Wu et al., 2006*). Moreover, natural odor plumes are encountered in various spatiotemporal patterns, and their dynamics and statistics can influence the neural encoding mechanism (*Brenner et al., 2000*; *Vickers et al., 2001*).

In *Drosophila*, OSNs expressing the same receptors connect with PNs in one of roughly 50 spherical compartments, termed olfactory glomeruli, which constitute a deutocerebral neuropil called

**eLife digest** Fruit flies are attracted to the smell of rotting fruit, and use it to guide them to nearby food sources. However, this task is made more challenging by the fact that the distribution of scent or odor molecules in the air is constantly changing. Fruit flies therefore need to cope with, and exploit, this variation if they are to use odors as cues.

Odor molecules bind to receptors on the surface of nerve cells called olfactory sensory neurons, and trigger nerve impulses that travel along these cells. The olfactory sensory neurons are connected to other cells called projection neurons that in turn relay information to the higher centers of the brain. While many studies have investigated how fruit flies can distinguish between different odors, less is known about how animals can use variation in the strength of an odor to guide them towards its source.

Kim et al. have now addressed this question by devising a method for delivering precise quantities of odors in controlled patterns to fruit flies, and then measuring the responses of olfactory sensory neurons and projection neurons. These experiments revealed that olfactory sensory neurons—which are found mainly in the flies' antennae—responded most strongly whenever an odor changed rapidly in strength, and showed relatively little response to constant odors. An independent study by Schulze, Gomez-Marin et al. found that olfactory sensory neurons in fruit fly larvae also respond in a similar way.

Kim et al. also found that the response of the projection neurons depended on both the rate of nerve impulses in the olfactory sensory neurons and on how quickly this rate was changing. But, unlike the olfactory sensory neurons, projection neurons showed their strongest responses immediately after an odor first appeared.

Thus, in contrast to organisms such as bacteria and worms, which are highly sensitive to the local concentration gradients of odors, fruit flies instead appear to be more responsive to the sudden appearance of an odor in their environment. Kim et al. suggest that this difference may reflect the fact that for ground-based organisms, local gradients are generally reliable predictors of the location of an odor source. However, for flying insects, continually changing air currents mean that predictable local gradients are less common. Therefore, the ability to detect a hint of an odor before the wind changes is a more useful skill.

the antennal lobe. PNs subsequently relay olfactory information to higher brain centers such as the mushroom body and the lateral horn (*Stocker et al., 1990*).

Two recent studies independently reported that *Drosophila* OSNs encode not only the odor concentration but also its rate of change as a function of time (*Kim et al., 2011*; *Nagel and Wilson, 2011*). Building on this recent advance, we asked how PNs further contribute to creating internal representations of dynamic olfactory environments. We tested OSNs and PNs with short plume-like odor stimuli in a variety of settings and analyzed the correlation structure of input/output signals in the odor-OSN-PN pathway. We also constructed a two-dimensional (2D) linear–nonlinear (LN) model of the OSN-to-PN transformation by inducing an ensemble of triangle-shaped OSN spike rates via a systematic design of olfactory stimuli.

## Results

We employed a novel odor delivery system that can reliably produce various odor concentration waveforms and provide measurements of the odor concentration with a millisecond resolution on every experiment trial (*Figure 1A,B*) (*Kim et al., 2011*). Various odor concentration profiles were designed and tested (*Figure 1—figure supplement 1*), and the corresponding OSN and PN responses were measured in two separate assays sharing the same odor delivery system (*Figure 1A,B*). The observed odor concentrations were closely matched between the two assays (*Figure 2A–C*). We used acetone as the primary odorant because its low ionization potential afforded a high signal-to-noise ratio in our odor concentration measurements. We tested a pair of directly connected OSNs and PNs innervating the DM4 glomerulus with five different acetone concentration waveforms. The dynamics of OSN and PN responses differed significantly from their respective feedforward inputs, and all responses initiated within a few tens of milliseconds of the odor onset (*Figure 1C*). PNs generally showed a bigger peak

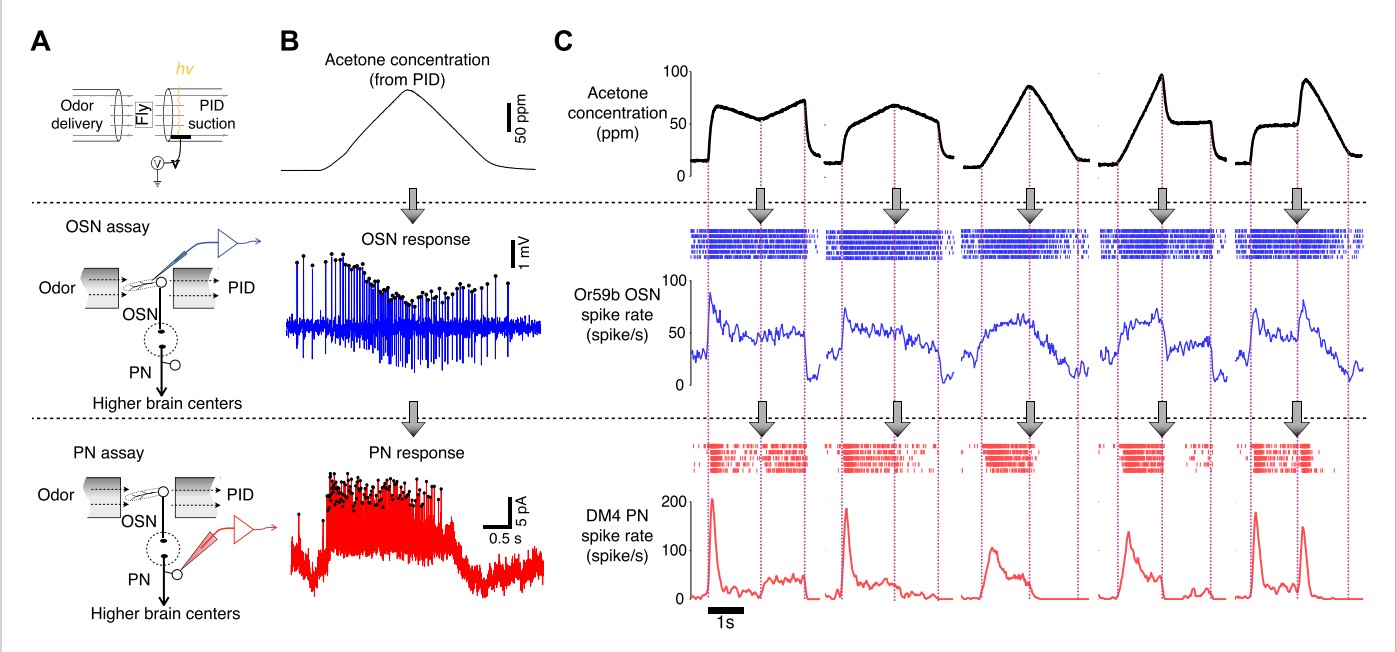

**Figure 1**. Dynamics of sample odor stimuli are significantly transformed along an odor-OSN-PN pathway. (**A**) An experimental setup. Activity of OSNs and PNs was recorded in two different assays, which share the same odor delivery system. A photoionization detector (PID) provided real-time measurements of odor concentrations in every trial. (**B**) Sample traces of OSN and PN responses to a triangle-shaped odor concentration profile. (**C**) Sample OSN and PN responses to five distinct odor concentration waveforms. (Top row) Odor concentration profiles. Each trace is an average of six interleaved trials, recorded in the OSN assay. (Middle row) Raster and peristimulus–time histogram (PSTH) plots of the Or59b OSN response. (Bottom row) Raster and PSTH plots of the postsynaptic DM4 PN response to the same panel of odor stimuli.

The following figure supplement is available for figure 1:

**Figure supplement 1**. Sample traces of 17 acetone odor concentration waveforms and their responses in Or59b OSNs and DM4 PNs.

spike rate and exhibited more phasic spiking patterns than the presynaptic OSNs. However, the exact functional transformation between OSNs and PNs could not be readily assessed due to the complex dynamics of OSN and PN signals.

We therefore designed a set of elementary odor concentration waveforms: a step, a ramp and a parabola (top row of *Figure 2A–C*). We reasoned that the simple nature of these waveforms would facilitate the analysis of the input/output relationship.

Consistent with previous reports (*Kim et al., 2011*; *Nagel and Wilson, 2011*), OSNs responded most strongly to polynomial waveforms when the odor concentration rose rapidly. For the ramp and parabola odor signals, OSN responses were the exact rate-of-change function of their input: a step output to a ramp input (*Figure 2B,E*) and a ramp output to a parabola input (*Figure 2C,F*). This pattern of encoding was preserved for different combinations of odorants and OSN-PN pairs (*Figure 2—figure supplement 1*). To look into this observation formally, we measured the similarity between three different input features—concentration amplitude, rate of change, and acceleration—and the output spike rate of OSNs by performing a set of cross-correlation analyses. The correlation was strong between the OSN output and both odor concentration and its rate of change, but weak between the OSN output and the acceleration of odor concentration (*Figure 2J*).

Given the very same panel of olfactory stimuli, PN responses exhibited bigger peak amplitude and more stereotyped temporal patterns than OSNs (*Figure 2G–I*). All PN responses transiently peaked at the time of the odor onset, and their peak times were on average advanced from the OSN peak times. Specifically, in response to ramp-shaped OSN signals, PNs exhibited a rapid step response (*Figure 2F,I*). To step-like OSN signals, PN showed a phasic onset response, followed by a tonic spiking pattern (*Figure 2E,H*). These input/output relationships are reminiscent of the rate-of-change

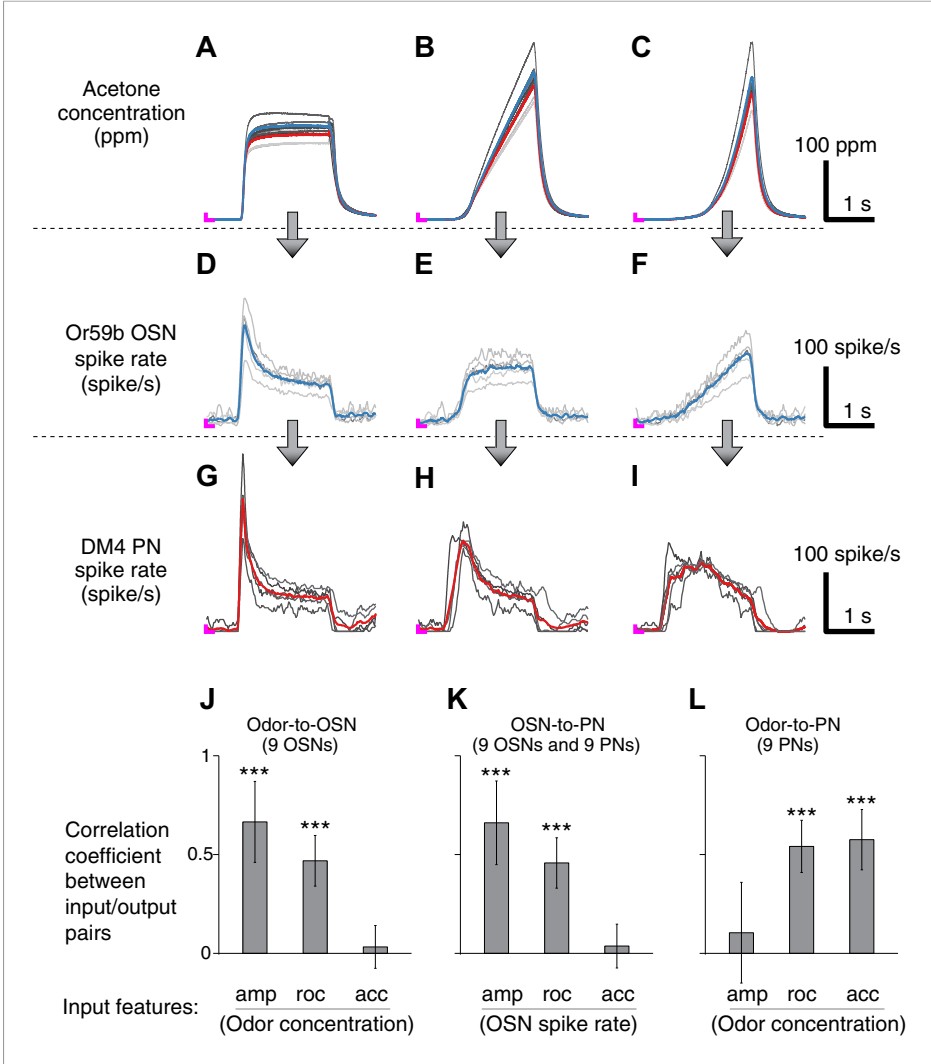

**Figure 2**. Correlation structures of olfactory information representations in odor, OSN and PN signals. (**A**–**C**) Three polynomial odor stimuli: a pulse, a ramp and a parabola. Light gray lines represent individual trials from five different OSN experiments, and dark gray lines represent individual traces from five different PN experiments. Blue and red lines are average traces, respectively, from the OSN and PN experiments. An 'L' mark in magenta at the bottom left corner of each panel represents the zero amplitude point. (**D**–**F**) Or59b OSN response to the above stimuli (n = 5 flies). (**G**–**I**) PN response to the same set of stimuli (n = 5 flies). (**J**–**L**) Correlation analyses between three pairs of input and output (amp: amplitude, roc: rate of change, acc: acceleration). OSNs and PNs mainly encode the amplitude and rate of change of their feedforward inputs, whereas PNs most strongly represent the acceleration and rate-of-change components of the odor input. Results with error bars indicate mean ± standard deviation, and ***indicates p < 0.001 (t-test). n = 9 flies for each analysis, 5 flies from the above traces and 4 flies from the same experiment at half concentration (data not shown).

The following figure supplement is available for figure 2:

**Figure supplement 1**. The patterns of the dynamic odor encoding were preserved for different combinations of odorants and OSN-PN pairs.

encoding in the odor-to-OSN transformation, where the ramp and step odor signals were similarly transformed. Biophysically, this phenomenon suggests a rapid adaptation between OSNs and PNs, which is in agreement with previous reports demonstrating strong short-term depression in synapses between OSNs and PNs (*Kazama and Wilson, 2008*, *2009*; *Nagel et al., 2015*). From a modeling perspective, this observation predicted a strong correlation between the PN spike rate and the rate of

change of the OSN spike rate. A correlation analysis between the PN output and three different input features—OSN spike rate, rate of change, and acceleration—confirmed this prediction (*Figure 2K*). Together, this supports the hypothesis that PN responses can be modeled as a function of the OSN spike rate and its rate of change, at least for the tested class of odor stimuli.

We therefore propose a simple model of the OSN-to-PN transformation comprising two input blocks: the OSN spike rate and its rate of change, followed by a 2D nonlinearity that maps these inputs into the PN spike rate (*Figure 3D*). The model has the structure of a classical 2D LN model with its linear blocks postulated as amplitude and rate-of-change filters (*Brenner et al., 2000*; *Geffen et al., 2009*). Given the hypothesized input blocks, the model can be fully identified by estimating the nonlinear block from experimental data. The validity of the model can be tested by assessing its ability to predict the PN response to new odor waveforms. In order to estimate the nonlinearity, we first designed OSN spike activity profiles that explore the hypothesized PN input space with an efficient

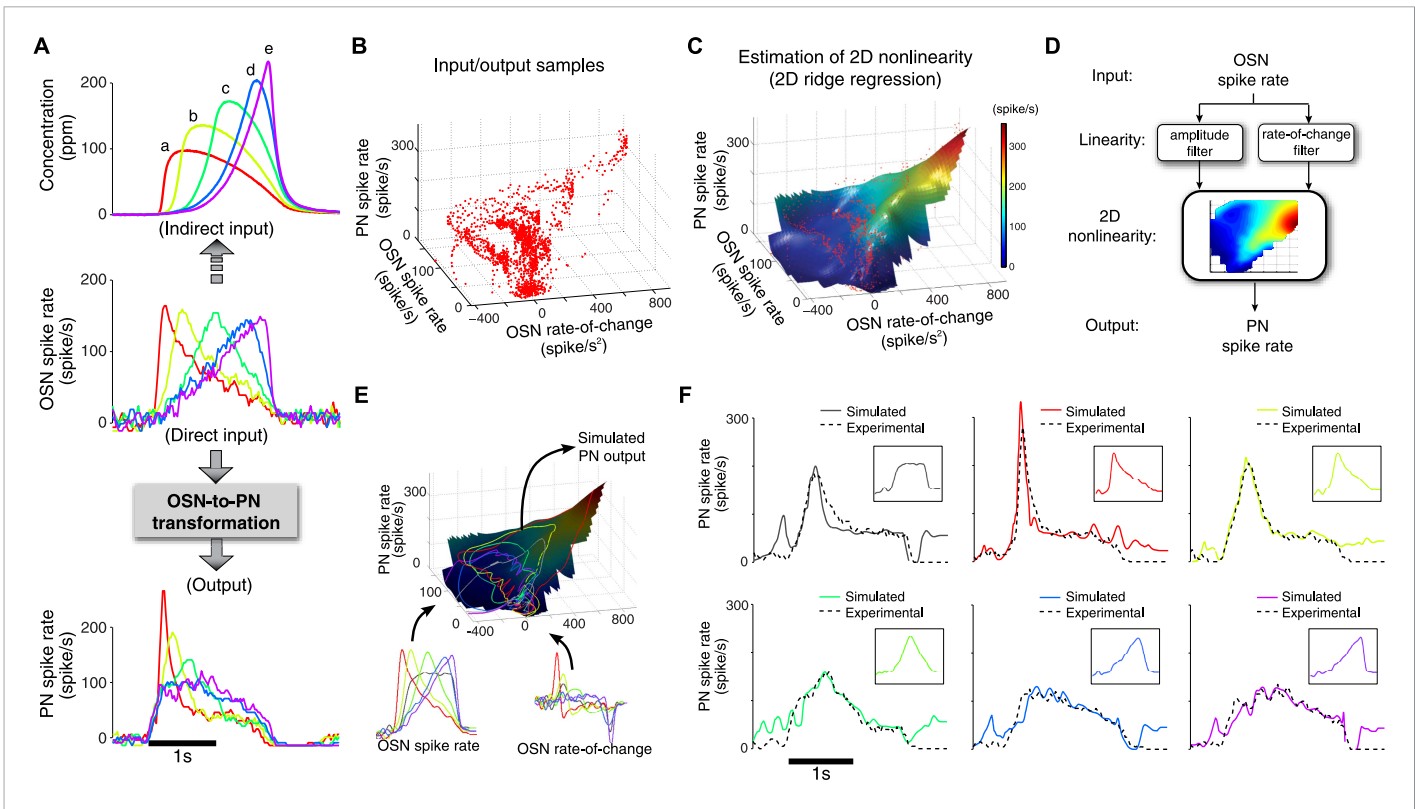

**Figure 3**. A simple two-dimensional (2D) model characterizes the OSN-to-PN transformation. (**A**) Input/output traces used in modeling the OSN-to-PN transformation (n = 5 trials for both OSN and PN data). An ensemble of triangle-shaped OSN spike rates (middle) were designed as a direct in vivo input to the PNs. These OSN spiking patterns were induced by an ensemble of parabola odor inputs (top). (**B**) Input and output samples for the 2D nonlinearity block. Input axes are OSN spike rate and its rate of change, and the output is the PN spike rate. The samples were estimated from the OSN and PN spike trains with 25-ms sampling interval and depicted as a red dot in the input/output space. (**C**) A 2D nonlinearity was estimated by ridge regression method from the samples marked in (**B**). (**D**) A simple 2D model of the OSN-to-PN transformation, as a 2D linear-nonlinear model. (**E**) The 2D model was tested for a set of OSN spike rates to evaluate its predictive ability. The black surface depicts a three-dimensional rendering of the 2D nonlinearity in (**B**). Six OSN spike rates—five triangle-shaped signals and one step-pulse-shaped input—and their rate-of-change functions were projected to the surface. Amplitude readouts from the trajectories on the surface constitute the simulated PN output. The step-pulse-shaped input was introduced for cross-validation of the model since this input was not used for building the model. (**F**) The predicted PN spike rate closely followed the magnitude and dynamics of the experimental PN spike rate, for six OSN spike rate inputs (insets). The average prediction errors are 31, 26, 21, 21, 23, 29 spike/s (clockwise from the top-left corner, in root-mean-square error).

The following figure supplement is available for figure 3:

**Figure supplement 1**. PN output is dependent on both OSN spike rate and its rate of change.

sampling grid. An ensemble of triangle-shaped inputs was shown to be suitable for this purpose, since each up/down ramp stimulus tests the target system with a distinct pair of positive and negative gradients while also sweeping a wide range of input amplitudes (*Kim et al., 2011*).

How can we induce an ensemble of triangle signals at the OSN output? We fine-tuned odor concentration signals, while observing the OSN spike rate, until the desired OSN spike rates were produced (*Figure 3A*). The odor concentration signals inferred from this method are an ensemble of parabolas with systematically varying peak times and amplitudes (*Figure 3A*). The 2D nonlinearity was estimated subsequently by running a ridge regression analysis on spike rate samples from the OSN and PN signals (*Kim et al., 2011*) (*Figure 3B,C*).

The estimated 2D nonlinearity showed a strong dependency on the rate-of-change of OSN spike rate (*Figure 3B,C*). When the nonlinearity was estimated for samples with a fixed odor concentration value, the PN spike rate rose almost linearly with respect to the rate of change (*Figure 3—figure supplement 1*). However, the gain of the PN output with respect to the OSN rate of change decreased monotonically with concentration, suggesting that at least a 2D model is required to describe the OSN-to-PN transformation (*Figure 3—figure supplement 1*). We tested the model by comparing the simulated model output with the experimental PN spike activity (*Figure 3C*). With step, ramp, and parabola odor inputs, the experimental outputs were closely matched by the model (root-mean-square error = 25 spike/s, *Figure 3F*). While the odor-evoked responses were well matched by the model, including peak times and amplitudes, the model output often overestimated the actual PN spike rate before and after the stimulus interval. This is because the estimated nonlinearity exhibits a relatively high slope at low input amplitudes (*Figure 3—figure supplement 1*), which renders the system highly sensitive to the noisy fluctuations at low OSN spike rates (*Pahlberg and Sampath, 2011*). This problem is thought to be mitigated in olfactory glomeruli by pooling input from a population of OSNs, thereby achieving a higher signal-to-noise ratio than provided by a single OSN (*Bhandawat et al., 2007*). In summary, the proposed simple 2D model provides a decent approximation of the OSN-to-PN transformation and corroborates the notion that PNs mainly encode OSN spike rate and its rate of change in a nonlinear fashion.

What is the functional consequence of acceleration encoding by antennal lobe PNs? In signal processing theory, the time derivative, or rate-of-change operation, advances the phase of a time signal. For example, $\cos(t)$ is the time derivative of $\sin(t)$, and its phase is advanced by $\pi/2$. Similarly, the rate of change of a sample triangle-shaped signal shows an advancement of a peak time, and its acceleration exhibits a peak time that is further advanced (*Figure 4A*). Therefore, we reasoned that the computation of the first and second time derivatives (rate of change and acceleration) of odor concentrations by OSN and PNs, respectively, acts to advance peak times of olfactory stimuli. To investigate this idea with a larger set of odor stimuli, we designed a set of triangle-shaped odor waveforms with peak times varying uniformly between 0.6 s and 1.7 s after the stimulus onset (*Figure 4B*, 'odor' curve in *Figure 4C*). As shown in the previous experiments with triangle-shaped stimuli (*Figure 2B*), OSNs responded in a pulse-shaped pattern (*Figure 4B*), with the peak response advanced by 400–1000 ms relative to the stimulus peak ('OSN' curve in *Figure 4C*). In response to the same panel of odor stimuli, PNs consistently produced peak output (*Figure 4B*) at around 200 ms after the stimulus onset, regardless of the dynamics of the odor/OSN signals ('PN' curve in *Figure 4C*). For the parabola-shaped odor stimuli (*Figure 3A*), the peak OSN and PN responses were similarly advanced in time relative to the odor and OSN peaks (right plot in *Figure 4C*) but were more variable. The advancement of a peak time was previously reported for step-pulse odor stimuli (*Bhandawat et al., 2007*), and our work confirms this result for a larger set of dynamically varying odor stimuli. Together, we hypothesize that acceleration encoding in the *Drosophila* antennal lobe allows higher olfactory centers to rapidly respond to onsets of slowly rising odor stimuli.

## Discussion

We tested first- and second-order neurons in the *Drosophila* olfactory system with a wide range of dynamically varying odor concentrations and constructed a simple 2D LN model for the temporal processing of olfactory information. The shape of the 2D nonlinearity and the relatively low prediction error of the model support the hypothesis that the PN spike rate is most strongly dependent on the rate of change of its feedforward OSN input. When combined with the dynamic sensory encoding by OSNs—encoding of odor concentration and its rate of change—PNs signal the rate of change and, most strongly, the acceleration of odor concentration signals to higher brain centers. One remarkable

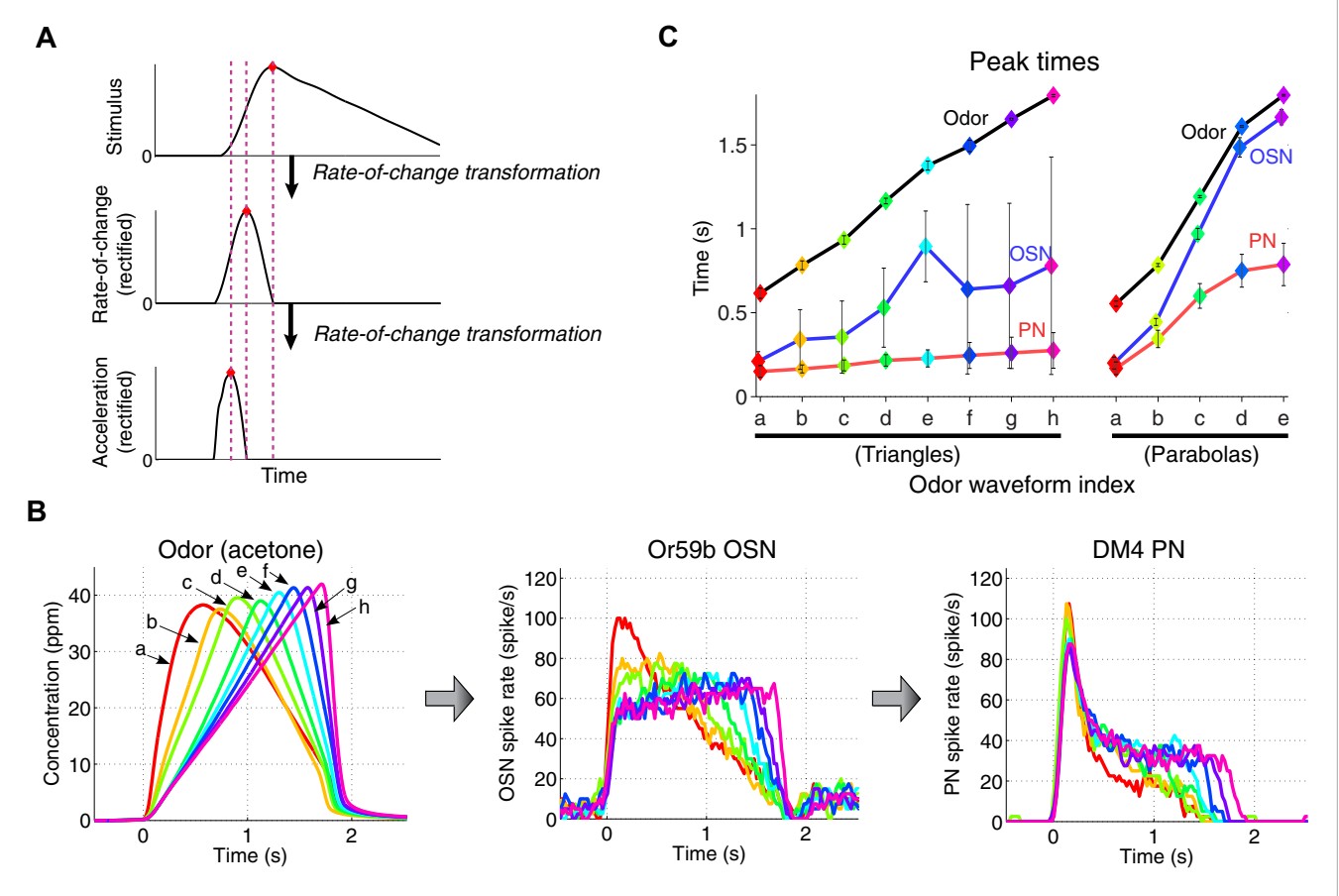

**Figure 4**. A cascade of rate-of-change encodings leads to a rapid detection of the stimulus onset. (**A**) A cascade of two rate-of-change transformations leads to the advancement of the peak time near to the stimulus onset, in response to a triangle-shaped input. At each stage, the rate of change of the input was computed, and the output was half-wave rectified. (**B**) 8 triangle-shaped acetone concentration waveforms (left) and the corresponding OSN and PN responses. (**C**) The peak times for odor, OSN, and PN signals. The advancement in peak times was observed in both the odor-to-OSN and OSN-to-PN transformations. Peak times were color coded so that the color of the marker matches that of the associated triangle signals in (**B**) and parabola signals in *Figure 3* (**A**). n = 5 OSNs and n = 4 PNs for both triangles and parabolas. Results with error bars indicate mean ± standard deviation.

example of the 'acceleration encoding' by PNs is found in the OSN/PN responses to the parabolically rising odor input, where the onset is enormously amplified to create step-like responses in PNs (*Figure 2C,I*).

To the best of our knowledge, the encoding of acceleration by an olfactory sensory system has not been demonstrated elsewhere. This sensory encoding mechanism is distinguished from other chemosensory systems. For example, in bacterial chemotaxis, the rate of change of a stimulus is computed and fed directly into its motor system that controls the flagellar rotation (*Bourret et al., 1991*). The nematode *Caenorhabditis elegans* also exhibits a local-gradient-based chemotaxis behavior, and neurons in its sensorimotor pathway are shown to compute the rate of change of concentration gradients (*Suzuki et al., 2008*).

Why do fruit flies compute acceleration of odor concentrations? How is the acceleration of an olfactory stimulus relevant to a fly's chemotaxis? An efficient chemotaxis strategy should depend on the distribution of odor molecules in the behaviorally relevant range, which can be quantified using a Reynolds number (*Weissburg, 2000*). Habitats of both nematodes and bacteria have low Reynolds numbers (Re < 1), and thus the odor distribution is primarily determined by a diffusion process. In this environment, the steepest local odor gradient can be directly linked to the location of the odor source, and therefore animals in such environments are expected to encode the rate of change of odor concentrations in order to detect the steepest gradient. In contrast, flying insects experience

fluid mechanics that have relatively high Reynolds numbers (Re > 10), with turbulence dictating the distribution of odor molecules over the diffusion process. In this regime, the local odor gradient within an odor plume would be less informative in determining the location of the odor source than the distribution of odor plumes over a macroscopic scale. In fact, it has been well established that insects, including fruit flies, use a strategy of turning upwind during odor plume encounters, a strategy that does not require interpreting local concentration gradients (*Budick and Dickinson, 2006*).

Furthermore, the relative position of a flying animal and the distribution of odor plumes evolve rapidly over time. It is therefore crucial for flying insects to detect and respond to an encountered odor plume before the fly's position drifts relative to adjacent plumes. We showed that *Drosophila*'s early olfactory system signals the acceleration of odor concentrations to higher brain centers. This encoding mechanism allows an animal to detect the odor onset at a very early phase (*Figure 4*). Therefore, it is tempting to speculate that the acceleration encoding has evolved in *Drosophila* to maximize its chance to locate the odor source in a turbulent environment.

It has also been shown that fruit flies can respond to the spatial difference of the odor concentration between two antennae in both walking and flight (*Duistermars et al., 2009*; *Gaudry et al., 2013*). However, even with the lack of spatial sampling ability, fruit fly larvae can locate an odor source, based on the temporal sampling of odor gradients (*Louis et al., 2008*). Therefore, adult fruit flies may rely on both spatial and temporal sampling methods to locate an odor source, although their relative importance may vary depending on the locomotive state, as Reynolds numbers are different between walking and flight.

## Materials and methods

### Odor delivery

We built a precise and versatile odor delivery system (*Kim et al., 2011*). An odorant was diluted in dipropylene glycol, and 20 ml of the resulting mixture was put into a glass vial (30 ml) and subsequently sealed by a screw cap with a silicone septa. Inlet and outlet needles were inserted into the vial headspace. The saturated vapor in this headspace was puffed by directing a low-flow air stream (≤100 ml/min) from an air cylinder (UN1002; TechAir, White Plains, NY) into the odor vial by opening inlet and outlet solenoid valves (Series 10; Parker-Hannifin Corporation, Cleveland, OH). Subsequently, the outlet needle carried the saturated vapor from inside the vial to a custom laminar mixer where the odor flow was combined with a strong carrier flow (800 ml/min). The mixer was carefully designed so as to minimize the turbulence that could be formed when the low-flow odor stream is added to a high-flow-rate carrier stream. The output of the mixer was then directed to the antennae and maxillary palps of a prepared animal through a glass capillary (1 mm inner diameter). The interval between two consecutive puffs was between 45 s and 70 s depending on the amount of vapor consumed in each puff, and the mixture was replenished about every 10 trials. All flows were electronically controlled by a miniature pressure controller (VSO-EP; Parker-Hannifin, OH) and measured using a digital flow meter (PV9000; Key Instruments, Trevose, PA) in real time. After hitting the animal, the odor flow was immediately sucked into a probe and measured by a photoionization detector (PID; Aurora Scientific, Canada) with 1 l/min flow rate. The output of the PID was digitally sampled at 10 kHz and later advanced by 2 ms in order to compensate for the latency caused by the suction probe. In order to convert the PID output voltage into the odor concentration in parts per million (ppm), we regularly (every 10 trials) measured the output of the PID with a calibration odor (3% propylene) and estimated the general sensitivity of the PID, which gradually decays over time. The PID sensitivity ratio of actual odor to standard odor was measured in a separate experiment with a fully saturated odor vapor and used to calculate a scaling factor between the PID output (in Volts) and the absolute odor concentration (in ppm) (*Kim et al., 2011*).

### *Drosophila* stocks

*Drosophila* stocks were maintained at room temperature on a 12-hr light/12-hr dark schedule and kept in standard plastic vials containing a cornmeal agar medium. All experiments were carried out with female flies of the *NP3062-GAL4/NP3062-GAL4;UAS-mCD8::GFP/UAS-mCD8::GFP;+/+* genotype. *NP3062-GAL* flies were kindly provided by Kei Ito.

## Electrophysiology

Female flies 2–5 days posteclosion were used for both OSN and PN recordings. For the OSN recordings, single basiconic sensilla were targeted with electrolytically sharpened tungsten electrodes to measure the aggregate electrical activity of OSNs housed in these sensilla (*Figure 1B*) (*Kim et al., 2011*). Recordings were analyzed by a custom MATLAB code (*Source code 1*) to sort out action potentials from Or59b or Or7a OSNs (*Kim et al., 2011*). For the PN recordings, a female fly was gently immobilized beneath a cellulose acetate film with an adhesive (*Datta et al., 2008*). The film was then carefully transferred to a petri dish in which a circular hole was made at the center. When placing an animal in the hole, the location of the head was carefully adjusted so that the antennae and maxillary pals were located strictly within the odor stream between the odor delivery tubing and the suction probe, both of which were embedded in the petri dish. Antennal lobes were exposed by gently removing a head cuticle as well as the film under saline solution containing 108 mM NaCl, 5 mM KCl, 2 mM $CaCl_2$, 8.2 mM $MgCl_2$, 4 mM $NaHCO_3$, 1 mM $NaH_2PO_4$, 5 mM trehalose, 10 mM sucrose, and 5 mM HEPES (pH, 7.5; 265 mOsm) (*Wang et al., 2003*). In order to access the PN somas, the perineural sheath was weakened by a 1–2 min treatment with 1 mg/ml collagenase (Type I Collagenase; Sigma–Aldrich, St. Louis, MO). The dorsal part of the sheath was then gently ruptured by a sharp quartz glass electrode using a micromanipulator (MP-285; Sutter Instrument, Novato, CA). A ground electrode was placed into the saline solution and connected to a headstage amplifier (CV-7B; Molecular Devices, Sunnyvale, CA).

In order to record action potentials of a single PN, a borosilicate patch electrode (7–10 MΩ tip resistance) filled with saline solution was attached to the target cell and gentle suction was applied through the electrode capillary until a loose seal (50–100 MΩ) was formed. The current was amplified (MultiClamp 700B; Molecular Devices, CA), low-pass filtered at 1.2 kHz, sampled at 10 kHz (Digidata 1322A; Axon Instruments), and stored on a computer by software (Clampex; Molecular Devices, CA). The animal was alive and active for a few hours as indicated by the spontaneous extension of its legs. Fresh oxygenated saline solution was perfused throughout the experiment. Finally, the PNs innervating the DM4 and DL5 glomeruli were identified by genetic means, using an enhancer trap line expressing Green Fluorescent Protein (GFP) in these cells. The identity of recorded cells was further verified by their odor response and the location of the soma in the dorsal cluster of the antennal lobe. Spike sequences were detected and sorted offline using custom software (*Source code 1*), written in MATLAB (MathWorks, Natick, MA).

## Estimating spike rates, rate of change, and acceleration signals

Spike rates of OSNs and PNs were estimated by constructing a peristimulus–time histogram (PSTH) with a 100-ms bin size and a 75-ms overlap between adjacent bins. For the correlation analysis and 2D model prediction, the rate of change and acceleration components of odor and OSN traces were estimated by a forward difference equation, with the difference time interval as the scaling factor; for example, $\frac{dx}{dt} = [x(t-\Delta) - x(t)]/\Delta$, where $x$ represents either the odor concentration or the OSN response, and $\Delta$ denotes the time interval. The time interval was 50 ms for the odor rate of change, 100 ms for odor acceleration, the OSN rate of change and OSN acceleration. Since the output of the forward difference equation is sensitive to high-frequency components of the signal, we low-pass filtered the amplitude and rate-of-change signals before deriving the rate of change and acceleration, respectively. Cut-off frequencies of the low-pass filters were 20 Hz for the odor amplitude, 10 Hz for the odor rate of change, 4 Hz for the OSN spike rate, and 4 Hz for the OSN rate of change.

## Cross-correlation analyses between input features and OSN/PN output

The cross-correlation coefficient, $\varrho(x,y)$ between an input $x$ and the corresponding output $y$ was computed as a normalized cross-covariance function; that is,

$$\varrho(x,y) = \frac{\frac{1}{N}\sum_{i=1}^{N}(x[i] - \mu_x)(y[i] - \mu_y)}{\sigma_x \sigma_y},$$

where $\mu$ and $\sigma$ are, respectively, the mean and standard deviation of the variable in the subscript.

## Estimating the 2D nonlinearity

We hypothesized a nonlinear function $f(.,.)$ that takes inputs from the OSN spike rate $y$ and its rate of change $dy/dt$ and whose output corresponds to the PN spike rate $z$; for example,

$$z = f\left(y, \frac{dy}{dt}\right),$$

and $f(.,.)$ was estimated using a 2D ridge regression method on $100 \times 100$ grids (*Kim et al., 2011*).

## Acknowledgements

The work presented here was supported by NIH under grant number R01DC008701-05 and was conducted in the Axel Laboratory at the Columbia University. The authors would like to thank Dr Richard Axel for his outstanding support and Vanessa Ruta, Jamie Fitzgerald, and Vikram Vijayan for helpful comments on an earlier version of the manuscript.

## Additional information

### Funding

| Funder | Grant reference | Author |
|---|---|---|
| National Institutes of Health (NIH) | R01DC008701-05 | Aurel A Lazar |

The funder had no role in study design, data collection and interpretation, or the decision to submit the work for publication.

### Author contributions

AJK, YBS, Conception and design, Acquisition of data, Analysis and interpretation of data, Drafting or revising the article, Contributed unpublished essential data or reagents; AAL, Conception and design, Analysis and interpretation of data, Drafting or revising the article

## Additional files

### Supplementary file

• Source code 1. Custom software written in MATLAB for the analysis of OSN and PN data.

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
