## [Decision Letter]

Thank you for sending your work entitled “Projection neurons in *Drosophila* antennal lobes signal the acceleration of odor concentrations” for consideration at *eLife*. Your article has been favorably evaluated by Eve Marder (Senior editor) and three reviewers, one of whom, Ronald L Calabrese, is a member of our Board of Reviewing Editors.

The Reviewing editor and the other reviewers discussed their comments before we reached this decision, and the Reviewing editor has assembled the following comments to help you prepare a revised submission.

The authors present a systematic electrophysiological analysis of the sensory transformation in the OSN (antennae) and PN (olfactory lobe) layers of the adult *Drosophila* olfactory system. They design clever and precise olfactory stimuli to dissect the dynamics of the responses of OSNs and PNs and develop a 2D model for the OSN to PN transformation. They conclude that OSNs respond to stimulus amplitude and rate of change while PNs respond to OSN spike rate and rate of change. Thus PN respond mainly to odor rate of change and acceleration. These results indicate that PNs enable flies to reliably respond to onsets of slowly rising odorant gradients.

The writing is clear and lively and the paper is nicely illustrated. The data provided are substantive and combined with insightful analyses and modeling argues strongly for the authors' conclusions. The Discussion is lively but does not place the work in a more general context of a fly navigating an odor plume and thus could be profitably expanded.

There were some concerns that must be addressed:

1) The novel finding here is with regard to the PN responses as previous work, albeit using less elegant stimuli, already characterized the dynamics of OSN activity to odor. Thus the Discussion should reflect this focus on the PNs.

2) The Linear-Nonlinear-Poisson model of Figure 3–figure supplement 2 caused considerable concern among the reviewers as the lack of fit at white noise onset was puzzling, causing them to wonder whether the technique has been properly applied. Moreover, the result is a negative one and does not impact upon the approach used by the authors. This analysis should be deleted and not discussed.

3) In revising the Discussion the authors should focus more on the implication for coding by PNs in more natural fluid conditions. What advantage does the second derivative coding of odor concentration confer to the detection or retention of an odor signal in flight through turbulent flow regimes that does not shift smoothly along polynomial trajectories?

4) The statistical concerns of Reviewer 2 should be thoroughly addressed.

These results have important implications for olfactory processing of realistic odor plumes in animals and indeed for sensory processing in general. Moreover, the results here have important behavior implications; they will be important in understanding chemotaxis.

*Minor comments*:

1) Figure 2: there seems to be an alignment problem. I suspect that the magenta “L” has been concatenated with the response line plots in A-C, but has been aligned with the response line plots in D-F, the effect of which is that the stimulus appears to lag *behind* the response when viewed along the vertical axis.

2) Although I do not myself need to see it, the authors are likely going to encounter some resistance to the profound lack of p-values. Statistical tests are frequently misused, but the authors could consider whether indicating mean correlation coefficients in the absence of some sort of statistic invites criticism.

3) In Figure 3, the simulations show a lot of noise at low odor concentration. This is a liability of any system sensitive to the first temporal derivative of stimulus intensity without some additional nonlinearity such as thresholding. This issue is examined in the visual system (Pahlberg and Sampath Bioessays 2011). Some discussion is warranted.

---

## [Author Response]

*1) The novel finding here is with regard to the PN responses as previous work, albeit using less elegant stimuli, already characterized the dynamics of OSN activity to odor. Thus the Discussion should reflect this focus on the PNs*.

While previous work has, to some extent, characterized the temporal processing in the OSN and PN layers in isolation from each other, an integrative view of early olfactory processing in *Drosophila* has been lacking. Our study characterizes the dynamic odor encoding in the early olfactory system as a whole. Furthermore, by employing precisely designed and measured time-varying olfactory stimuli, our study identifies specific patterns of dynamic olfactory information received by higher brain centers and provides a framework for predicting olfactory information representation in response to a given odorant concentration profile. Based on the comment, we have revised the Discussion section to place more emphasis on PNs and how their response may facilitate anemotaxis behavior in fruit flies.

*2) The Linear-Nonlinear-Poisson model of Figure 3–figure supplement 2 caused considerable concern among the reviewers as the lack of fit at white noise onset was puzzling, causing them to wonder whether the technique has been properly applied. Moreover, the result is a negative one and does not impact upon the approach used by the authors. This analysis should be deleted and not discussed*.

Figure 3–figure supplement 2 is dropped accordingly.

*3) In revising the Discussion the authors should focus more on the implication for coding by PNs in more natural fluid conditions*. *What advantage does the second derivative coding of odor concentration confer to the detection or retention of an odor signal in flight through turbulent flow regimes that does not shift smoothly along polynomial trajectories*?

Although it is speculative at this point, we believe that acceleration encoding found in our study has evolved to facilitate the plume-based anemotaxis. This is in contrast to the rate-of-change encoding employed by nematodes and bacteria in diffusion-dominant fluid environments.

*4) The statistical concerns below should be thoroughly addressed*.

We added statistical tests for the cross-correlation analysis in Figure 2.

Minor comments:

*1)*
Figure 2*: there seems to be an alignment problem. I suspect that the magenta “L” has been concatenated with the response line plots in A-C, but has been aligned with the response line plots in D-F, the effect of which is that the stimulus appears to lag* behind *the response when viewed along the vertical axis*.

We examined the alignment issue and found that ∼50 ms advancement of OSN/PN PSTH onsets relative to the odor onset. However, this is an artifact of computing PSTH out of spike sequences with 100 ms bin size. Because the 100 ms bin is essentially a step-pulse smoothing window, PSTHs can appear up to 50 ms earlier than the actual onset of spike outputs. We added this explanation to the revised caption of Figure 2. Because both the OSN and PN spike rates were estimated with the same bin size, the advancement artifact applies equally to both, and thus causality in the ONS-to-PN transformation is preserved.

*2) Although I do not myself need to see it, the authors are likely going to encounter some resistance to the profound lack of p-values. Statistical tests are frequently misused, but the authors could consider whether indicating mean correlation coefficients in the absence of some sort of statistic invites criticism*.

We have performed the student two-tailed *t*-test on the data in Figure 2 and found that the results were consistent with the description in our original manuscript. We have added these statistical test results in Figure 2.

*3) In*
Figure 3*, the simulations show a lot of noise at low odor concentration. This is a liability of any system sensitive to the first temporal derivative of stimulus intensity without some additional nonlinearity such as thresholding. This issue is examined in the visual system (Pahlberg and Sampath Bioessays 2011). Some discussion is warranted*.

We have extended the result paragraph to clearly explain this limitation of our model and how the actual PNs overcome this problem by pooling inputs from a population of OSNs with the same receptors.